# Sequentially amplified circularly polarized ultraviolet luminescence for enantioselective photopolymerization

Dongxue Han[1,2,4], Xuefeng Yang[2,3,4], Jianlei Han[2], Jin Zhou[2], Tifeng Jiao [1] & Pengfei Duan [2,3✉]

Chiral optical materials based on circularly polarized luminescence (CPL) have emerged rapidly due to their feasible applications in diverse fields of research. However, limited to the small luminescence dissymmetry factor ($g_{lum}$), real application examples have rarely been reported. Here, we present a complex system, which show intense circularly polarized ultraviolet luminescence (CPUVL) with large $g_{lum}$ value, enabling a chiral UV light triggered enantioselective polymerization. By integrating sensitized triplet-triplet annihilation upconversion and CPL, both visible-to-UV upconversion emission and upconverted circularly polarized ultraviolet luminescence (UC-CPUVL) were obtained in the systems, built of chiral annihilator $R(S)$-4,12-biphenyl[2,2]paracyclophane ($R$-/$S$-TP), and a thermally activated delayed fluorescence (TADF) sensitizer. After dispersing this upconversion system into room-temperature nematic liquid crystal, induced chiral nematic liquid crystal could significantly amplify the $g_{lum}$ value (0.19) of UC-CPUVL. Further, the UC-CPUVL emission has been used to trigger the enantioselective photopolymerization of diacetylene. This work paves the way for the further development of functional application of CPL active materials.

[1] State Key Laboratory of Metastable Materials Science and Technology, Yanshan University, No. 438 West Hebei Street, Qinhuangdao 066004, P.R. China. [2] CAS Center for Excellence in Nanoscience, CAS Key Laboratory of Nanosystem and Hierarchical Fabrication, National Center for Nanoscience and Technology (NCNST), No. 11 ZhongGuanCun BeiYiTiao, Beijing 100190, P.R. China. [3] University of Chinese Academy of Sciences, No.19 (A) Yuquan Road, Shijingshan District, Beijing 100049, China. [4] These authors contributed equally: Dongxue Han, Xuefeng Yang. ✉email: duanpf@nanoctr.cn

Circularly polarized luminescence (CPL) has gained considerable interest[1–12] in recent years thanks to their widely potential applications in optical displays[1,13–15], disease detection[16] and even as catalysts for asymmetric photochemical synthesis[17–19]. However, real applications have rarely been reported due to the small luminescence dissymmetry factor ($g_{lum}$). In terms of circularly polarized luminescent materials, the luminescence dissymmetry factor is defined as $g_{lum} = 2 \times (I_L - I_R)/(I_L + I_R)$, where $I_L$ and $I_R$ signifies respectively the left- and right-handed CPL intensity, are applied to quantify the level of CPL performance. Therefore, the $g_{lum}$ can vary from $-2$ to $+2$, which corresponds to a total emission of right- and left-handed circularly polarized light[20]. Searching large $g_{lum}$ value of CPL-active materials is a critical issue for real application in this field. Nevertheless, in organic systems, particularly chiral small organic molecules, the $g_{lum}$ value was not satisfying for practical application despite their good performance in other aspects. In this sense, over the past few years, various strategies for amplifying $g_{lum}$ value have been developed, including supramolecular self-assembly[21–24], aggregation-induced emission[11,25–28], energy transfer involved circularly polarized excitation[29–32], incorporating into chiral liquid crystals[33–35], and so forth. Nevertheless, the reported CPL-active systems with high $g_{lum}$ value were almost located in the visible light region, whose relatively low energy limited their practical applications. Accordingly, the development of circularly polarized ultraviolet luminescence (CPUVL) with high $g_{lum}$ value and further expanding its application remain a big challenge.

Recently, we have reported one kind of upconverted CPL (UC-CPL), which was constructed by integrating triplet–triplet annihilation upconversion (TTA-UC) and CPL[36]. The triplet–triplet energy transfer involved TTA-UC process could significantly amplify the circular polarization of the UC-CPL. Considering that TTA-UC can easily realize upconverted UV emission by flexible selecting appropriate sensitizer/annihilator combination, and the remarkable $g_{lum}$ value amplification effect of UC-CPL process, a

well-designed sensitizer/annihilator pair of TTA-UC, which possess excellent performance in the upconverted CPUVL (UC-CPUVL), is required.

Photon upconversion affords the possibility to convert low energy photons into high energy photons. The TTA-UC has an expansion to a wider range of scientific fields in terms of applications, such as photovoltaics[37–39], biomolecular systems[40], and photocatalysis[41,42] primarily due to its high efficiency, low-power excitation[37–47]. Undoubtedly, visible-to-UV upconversion is particularly important because many applications of upconversion deal with the presence of high energy. However, few examples of visible-to-UV TTA-UC can be found in the literatures and their upconversion efficiency is unsatisfactory because of the unsuitable energy levels between sensitizers and annihilators[45,48–51]. More importantly, the UC-CPUVL with high $g_{lum}$ value, to the best of our knowledge, remains unexploited.

Although the amplification of CPL emission through TTA-UC has been demonstrated to be an efficient approach, the amplified $g_{lum}$ values are still very small in a range from $10^{-4}$ to $10^{-3}$, which is pretty far from the requirement of real application. On the other hand, chiral nematic liquid crystal (N*LC) has been regarded as one kind of smart materials to realize CPL and UC-CPL with larger $g_{lum}$ value, attributing to its unique optical properties, such as optical rotation and circular dichroism (CD)[35,52–55]. Thus, by incorporating the CPL-active TTA-UC pairs into N*LC, the $g_{lum}$ value of UC-CPL should be remarkably boosted.

Here, we report an example of visible-to-UV UC-CPUVL, which is composed of chiral annihilator R(S)-4,12-biphenyl[2,2] paracyclophane (R-/S-TP) and achiral sensitizer thermally activated delayed fluorescence (TADF) molecule 4CzIPN[56]. It should be noted that, thanks to the excellent performance of TADF molecule in small $S_1$–$T_1$ gap ($\Delta E_{ST}$) and less energy loss during the intersystem crossing process[57], the combination shows a reasonably high photon upconversion efficiency ($\Phi_{UC} = 7.9\%$). Due to the essential planar chirality, the annihilator R-/S-TP

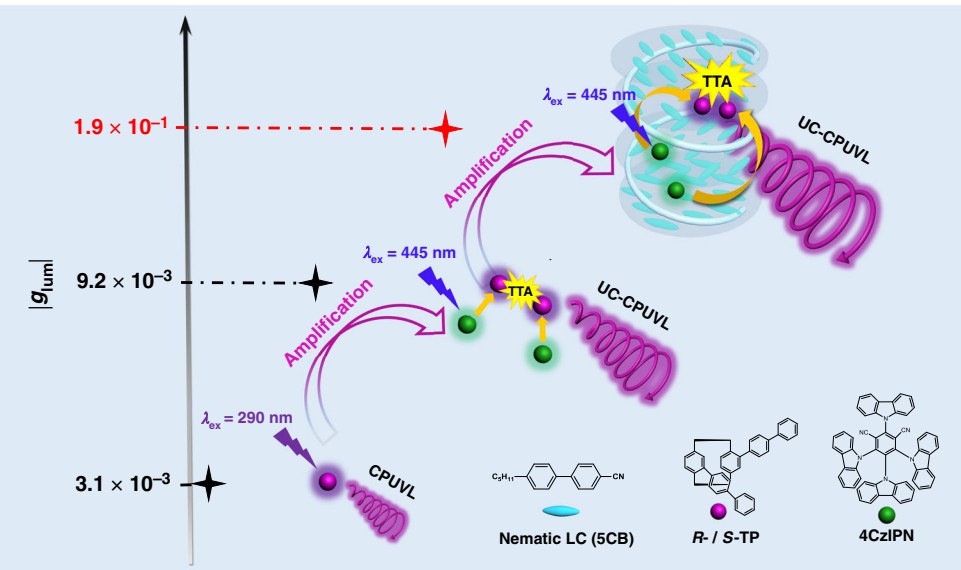

**Fig. 1 The process of sequentially amplified CPUVL.** Chemical structures of R-/S-TP, 4CzIPN, 5CB and schematic representation of sequentially amplified CPUVL. R(S)-TP showed a weak CPUVL emission ($|g_{lum}| = 3.1 \times 10^{-3}$) in toluene solution when excited by the 290 nm. After embedding with achiral sensitizer 4CzIPN, UC-CPUVL could be observed under the excitation of 445 nm continuous wave laser (A 445 nm short-pass filter was used). Due to the amplification effect photon upconversion on CPUVL emission, the $g_{lum}$ value of UC-CPUVL reached to $9.2 \times 10^{-3}$ and showed three times larger than the downconversion process. Subsequently, chiral emitter and TADF sensitizer were adopted into the nematic liquid crystal (5CB), induced N*LC performed enhanced UC-CPUVL with a larger $g_{lum}$ value ($|g_{lum}| = 1.9 \times 10^{-1}$), which could trigger the enantioselective polymerization of diacetylene with good stereoselectivity.

shows good CPL performance in dilute solution ($|g_{lum}| = 3.1 \times 10^{-3}$, Fig. 1). TTA-UC process can further amplify the circular polarization of UC-CPUVL to a larger value ($|g_{lum}| = 9.2 \times 10^{-3}$). In addition, after embedding this upconversion pairs into a room-temperature nematic liquid crystal, induced N*LC and high performance UC-CPUVL can be easily obtained. It's worth mentioning that the $g_{lum}$ value reaches to 0.19, which is almost three times larger than the prompt CPL in N*LC ($7.5 \times 10^{-2}$). This larger $g_{lum}$ value of upconverted UV light motivates us to put it into practical application. Here, taking a step forward, the UC-CPUVL emission generated from N*LC, has been successfully used to trigger the enantioselective polymerization of diacetylen with great stereoselectivity.

## Results

**Prompt CPUVL behavior in solution.** It is acknowledged that the combination of chiral group with organic emitter was one of the most direct and efficient routes to construct CPL-active materials. Thus, following this strategy, the employed chiral annihilator R(S)-TP, was synthesized directly R(S)-4,12-dibromo [2,2]paracyclophane and 1,4-biphenylboronic acid by Suzuki coupling reactions (Supplementary Methods and Supplementary Fig. 1). Subsequently, R-TP was chosen as an example for exploring the photophysical properties, and its absorption and emission spectra were shown in Fig. 2a, with a concentration of 0.01 mM in THF solution. It could be clearly observed that R-TP showed a characteristic vibrational structure of p-terphenyl. The absorption spectrum showed intense absorption band at 255 and 290 nm, and its maximum emission located at 380 nm under the excitation of 290 nm. Notably, compared with the p-terphenyl, both absorption and emission spectra of R-TP showed obvious red shift due to the increased π-conjugation length.

Subsequently, we explored the chiroptical properties of R(S)-TP through testing the CD and CPL spectra. As shown in Fig. 2b, R-TP and S-TP showed a mirror-image CD signals. R-TP possessed a positive Cotton effect at 330 nm and a negative Cotton effect at 271 nm. In addition, both enantiomers showed clear CPUVL emission at 380 nm, and its corresponding $g_{lum}$ value was about $3.1 \times 10^{-3}$. It should be noted that positive signal represented the R-TP exhibited left-handed CPL emission while negative signal corresponding to the opposite one, which demonstrated the achiral π-conjugated chromophores were endowed chiral emission after tethering the planar chirality[58].

**TTA-UC and UC-CPUVL behavior in solution.** To our knowledge, visible-to-UV TTA-UC, where p-terphenyl and 4CzIPN were utilized as annihilator and sensitizer, respectively, has been explored. However, in this work, better TTA-UC efficiency was obtained by replacing the p-terphenyl with chiral annihilator R(S)-TP. The solubility of R(S)-TP in toluene is better than in THF solution, which was beneficial to the TTET process in TTA-UC. Thus, we have thoroughly investigated the TTA-UC behavior in toluene solution (the experimental setup is presented in Supplementary Fig. 3). The photophysical properties of anni-hilators, as well as the TADF sensitizer in toluene solution have been examined (Supplementary Fig. 2). We have carefully tested the solubility of the annihilators in toluene, and the concentration of 3 mM was found to be the maximum concentration. To get high efficient TTA-UC, a larger concentration of annihilator is required. We then thoroughly tested the TTA-UC behavior by fixing the R-TP concentration at 3 mM, blending with various amount of sensitizer 4CzIPN. It has been confirmed that, when the mixing molar ratio of R-TP/4CzIPN at 3 mM/0.1 mM, we could observe the best TTA-UC emission in toluene solution. As shown in Fig. 3a, after fixing the concentration of R-TP and

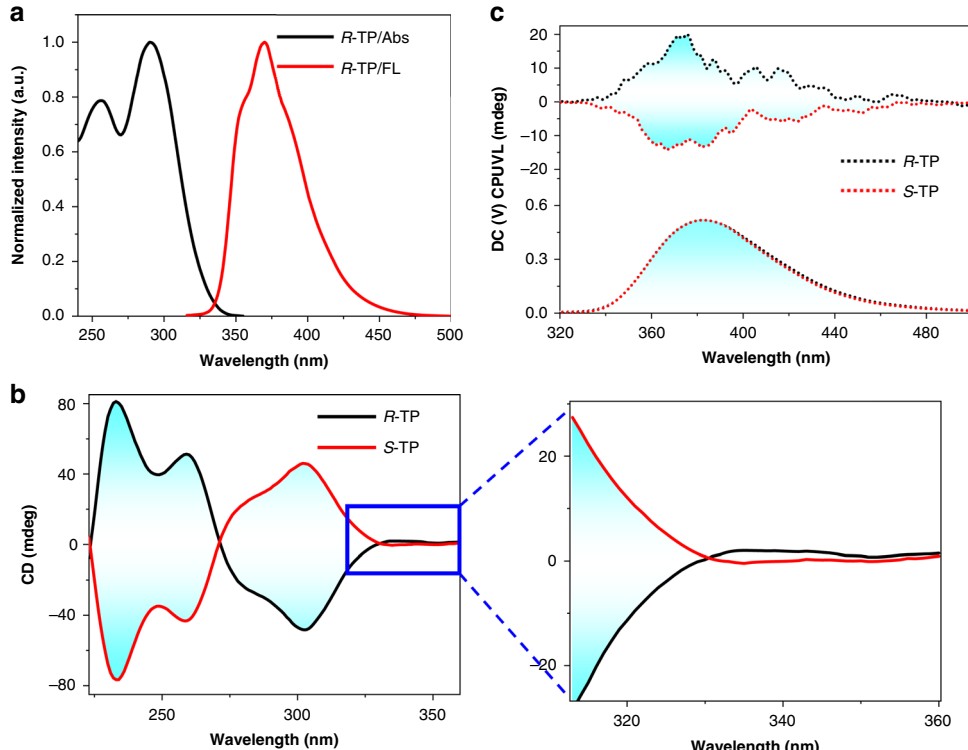

**Fig. 2 CD and CPUVL spectra of R-/S-TP in THF solution. a** Normalized absorption and emission spectra of R-TP (0.01 mM, $\lambda_{ex} = 290$ nm) in THF solution. **b** CD spectra of R(S)-TP in THF solution ([R-TP] = [S-TP] = 0.01 mM). **c** CPL spectra of R(S)-TP in THF solution ([R-TP] = [S-TP] = 0.01 mM, $\lambda_{ex} = 290$ nm).

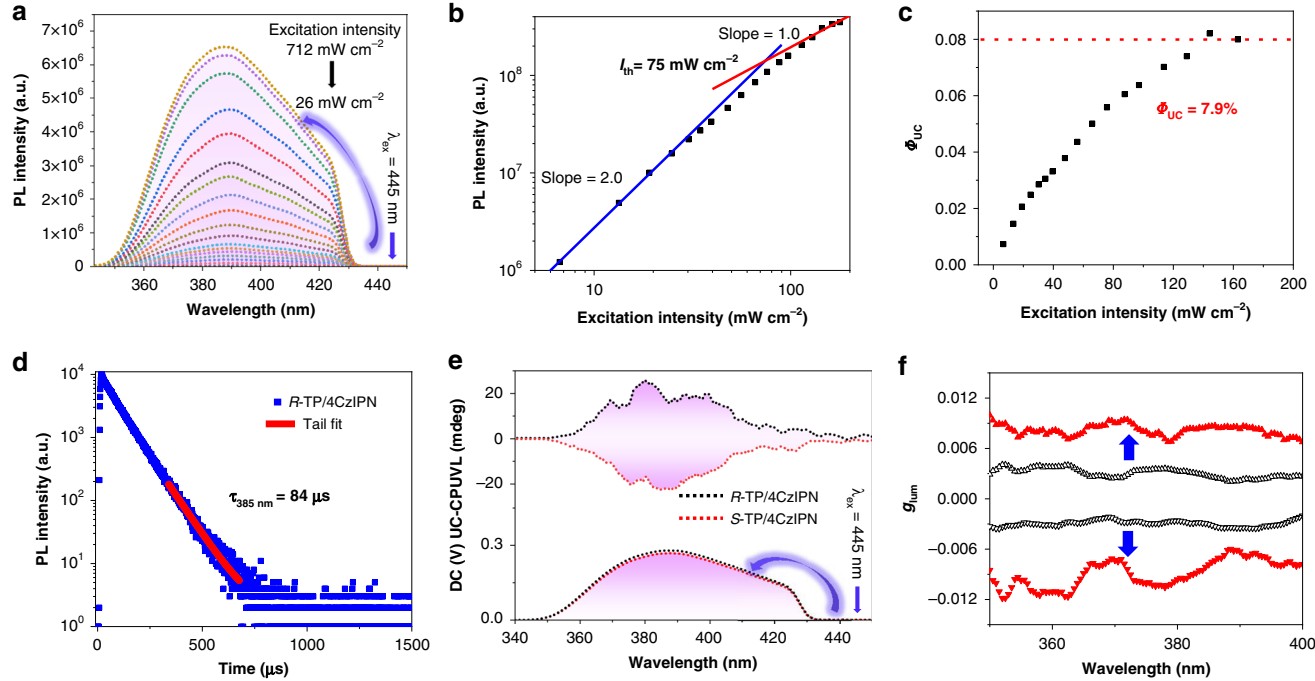

**Fig. 3 Upconversion and UC-CPUVL process of R(S)-TP/4CzIPN in deaerated toluene solution. a** Upconversion emission spectra of R-TP/4CzIPN with different incident excitation intensities of the 445 nm CW laser in deaerated toluene solution. **b** Double-logarithmic plots of the upconversion emission intensity of R-TP/4CzIPN in deaerated toluene solution as a function of the excitation intensity. **c** Upconversion efficiencies of R-TP/4CzIPN in deaerated toluene solution with different excitation light intensities of the 445 nm CW laser. **d** Time-resolved upconverted emission at 385 nm of the R-TP/4CzIPN in toluene solution ($\lambda_{ex}$ = 445 nm). **e** UC-CPUVL spectra of R-TP/4CzIPN and S-TP/4CzIPN in deaerated toluene solution ($\lambda_{ex}$ = 445 nm). **f** Luminescence dissymmetry factor $g_{lum}$ of UC-CPUVL versus wavelength. R-TP/4CzIPN excited by 445 nm CW laser (red triangle); R-TP excited by 290 nm (white triangle); S-TP/4CzIPN excited by 445 nm CW laser (red inverted triangle); S-TP excited by 290 nm (white inverted triangle). ([R-TP] = [S-TP] = 3 mM, [4CzIPN] = 0.1 mM). For all the upconversion measurements, a 445 nm short-pass filter was used.

4CzIPN in toluene solution at 3 and 0.1 mM, respectively, upconverted UV emission spectra with different incident power densities were obtained with the excitation of 445 nm continuous wave laser. The location of the upconversion emission is similar to the normal fluorescence ($\lambda_{ex}$ = 290 nm). In order to deeply understand the TTA-UC mechanism, the algebraic relationship between the upconversion emission intensity and the excitation power density was measured. Figure 3b presented the double-logarithmic plots of the upconverted UV emission intensity as a function of incident light power density. The blue and red lines (slopes = 2.0 and 1.0, respectively) showed the perfect fitting results in the range of low and high excitation intensity, which was the vitally important experimental evidence for typical TTA-UC system[59]. In addition, the threshold excitation intensity ($I_{th}$) was 75 mW cm$^{-2}$, where TTA became the main deactivation channel for the annihilator triplet and the upconversion quantum yield reached saturation. Figure 3c showed the dependence of the TTA-UC quantum efficiency $\Phi_{UC}$ on the excitation power intensity for the mixed solution of R-TP/4CzIPN, a saturation quantum yield of about 7.9% was obtained by using Coumarin 6 in DMF as a standard when the maximum efficiency is standardized to be 100% (Supplementary Methods)[60]. This value is relatively large compared with previous reports[45,48,50]. The TTA-UC mechanism could also be confirmed by the lifetime measurement. As shown in Fig. 3d, the upconverted UV emission lifetime at 385 nm was 84 μs, which could be ascribed to the mechanism based on long-lived triplet species. In addition, after comparing the lifetime of 4CzIPN in the TTA-UC process with its normal luminescence lifetime in solution, the emission decays of 4CzIPN decrease dramatically from 149 to 34 μs when the addition of annihilators (Supplementary Fig. 4), indicating a

sensitizer-to-annihilator fast and efficient TTET process. The corresponding TTA-UC measurements of S-TP/4CzIPN pair were collected in Supplementary Fig. 5. Accordingly, in the excitation of 445 nm CW laser, CPUVL based on the TTA-UC process, named UC-CPUVL, was successfully realized, whose emission was consistent with the prompt CPUVL of R(S)-TP (Fig. 3e). The corresponding $g_{lum}$ value of UC-CPUVL presented three times larger than the prompt CPUVL, from $3.1 \times 10^{-3}$ to $9.2 \times 10^{-3}$, which could be attributed to the chirality-induced spin polarization of singlet excitons resulting from the TTA-UC process (Fig. 3f)[61].

**CPUVL and UC-CPUVL behavior in N\*LC.** Currently, doping the chiral emitters into the achiral nematic liquid crystal was considered as a preferred method to realize CPL-active N\*LC. In this work, we showed the example of enhanced UC-CPUVL emission in N\*LC, which could be easily achieved by blending a chiral TTA-UC pair with an achiral room-temperature nematic liquid crystal (5CB). Firstly, before exploring the upconverted emission properties in the N\*LC, we separately tested the luminescence spectra of annihilator R-TP and sensitizer 4CzIPN in 5CB. As shown in Supplementary Fig. 6, compared to the monodisperse state, both the emission of R-TP and 4CzIPN in 5CB showed a 20 nm red shift. Normally, in a certain range, increasing the weight ratio of chiral molecule is favorable to the induction of chiral nematic liquid crystal. However, the properties of host liquid crystal (room-temperature nematic liquid crystal 5CB is used in this work) will perform significant changes, such as viscosity increase or clear point decrease, after adding an excess of chiral dopants. Thus, it is essential to thoroughly explore the CPL activity of various mixing weight ratios of chiral dopant. As

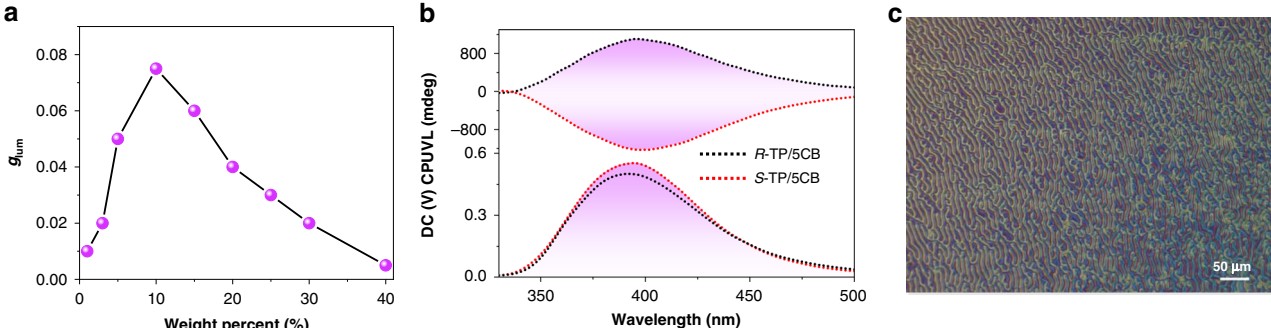

**Fig. 4 CPUVL behavior of R(S)-TP/4CzIPN in N*LC. a** The tendency of CPUVL luminescence dissymmetry factor $g_{lum}$ with different mixing weight ratios of R-TP/5CB ($\lambda_{ex}$ = 290 nm). **b** CPUVL spectra of R(S)-TP/5CB (10 wt%, $\lambda_{ex}$ = 290 nm, $|g_{lum}|$ = 7.3 × 10$^{-2}$). **c** Polarizing optical microscope image of R-TP/5CB, the mixing weight ratio is 10 wt%.

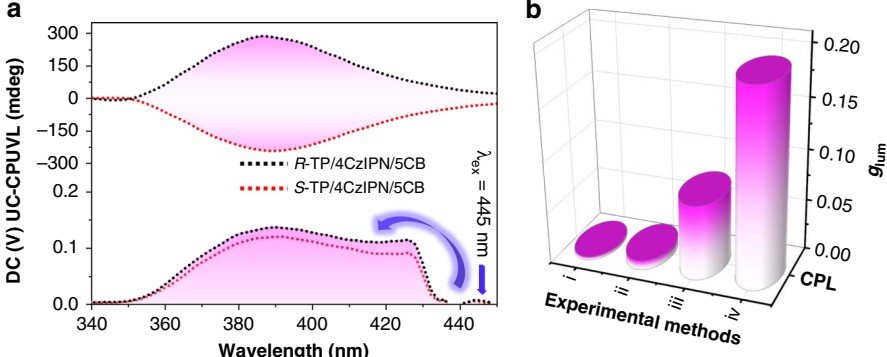

**Fig. 5 UC-CPUVL of R(S)-TP/4CzIPN in N*LC. a** UC-CPUVL spectra of R(S)-TP/4CzIPN in N*LC with incident excitation of the 445 nm CW laser (A 445 nm short-pass filter was used). **b** Collection of luminescence dissymmetry factor $g_{lum}$ of (i) R-TP in toluene solution excited at 290 nm ([R-TP] = 0.01 mM); (ii) R-TP/4CzIPN in toluene solution excited by 445 nm CW laser ([R-TP] = 3 mM, [4CzIPN] = 0.1 mM); (iii) R-TP in N*LC excited at 290 nm (R-TP/5CB = 10 wt%); (iv) R-TP/4CzIPN in N*LC excited by 445 nm CW laser (R-TP/5CB = 10 wt%, 4CzIPN/R-TP = 3 mol%).

shown in Fig. 4a, the highest $g_{lum}$ value is obtained when the mixing weight ratio of R-TP/5CB was 10 wt%, indicating that the chiral nematic liquid crystal possessed the most favorable configuration in this case. Since the CPL signal direction can be determined by the handedness of N*LC, it can be clearly clarified that R-TP will induce the 5CB to a right-handed N*LC, while the addition of S-TP to 5CB will result in a left-handed N*LC. Subsequently, we had also checked the polarizing optical microscope images of R-TP in 5CB (Fig. 4c). The typical fingerprint texture proved that the N*LC has been induced perfectly by the chiral annihilator. After embedding with 4CzIPN, the fingerprint texture could still be preserved (Supplementary Fig. 7). Additionally, after fixing the mixing weight ratio of R(S)-TP/5CB at 10 wt%, 4CzIPN could also exhibit the intense CPL emission in the N*LC, whose direction followed the regulation of N*LC (Supplementary Fig. 8). These results explicitly demonstrated that the CPL signal direction of emitters was determined by the handedness of N*LC.

Rather surprisingly, after blending with 4CzIPN, the UC-CPUVL with a larger $g_{lum}$ value ($|g_{lum}|$ = 1.9 × 10$^{-1}$) was achieved through the TTA-UC process in the N*LC. Similarly, the upconverted $g_{lum}$ exhibited the same tendency compared to the prompt CPUVL in 5CB. After fixing the mixing molar ratio of 4CzIPN/R-TP around 3 mol%, the $g_{lum}$ value came up to the highest point when the weight ratio of R-TP/5CB reached 10 wt% (Supplementary Fig. 9). As expected, a mirror-image UC-CPUVL signal could be observed under excitation by the 445 nm CW laser, whose $g_{lum}$ value was around 0.2 (Fig. 5a). It was in agreement with the prompt CPUVL in the N*LC. However, in this study, the intensity of TTA-UC emission in N*LC was

relatively weak ($\Phi_{UC}$ ~2.3%), even the excitation of 445 nm CW laser was in a high incident power density (Supplementary Fig. 10), which should be ascribed to the bad dispersion state of sensitizer/annihilator pair in N*LC. We have thoroughly compared the $g_{lum}$ values of all the tested CPL behavior (Fig. 5b). Obviously, in chiral liquid crystal, the $g_{lum}$ value of UC-CPUVL was almost three times magnitude of the prompt CPUVL. Impressively, comparing with the values in dilute solution, after incorporating into chiral liquid crystal, $g_{lum}$ value of CPUVL was magnified 60 times, while $g_{lum}$ value of UC-CPUVL was magnified 20 times.

**UC-CPUVL triggered enantioselective photopolymerization.** For the development of practical applications of the UC-CPUVL with higher $g_{lum}$ value (0.19) generated from the TTA-UC system, enantioselective photopolymerization would be a desirable application direction. Since the chiral UV light possesses high energy, it has been widely used for initiating the enantioselective photopolymerization of diacetylene[62–65]. In our previous work, we have reported one typical example about the enantioselective photopolymerization of diacetylene, which could be triggered by applying the upconverted circularly polarized light generated from a co-gel composed of upconversion nanoparticles (UCNPs) and chiral gelator[66]. However, since the obtained $g_{lum}$ value of CPL from the UCNPs-doped co-gel was small (~10$^{-3}$), the chiral direction of predominant helical polydiacetylene (PDA) chain could not be controlled very well. Thus, whether the TTA-UC process based UC-CPUVL with high $g_{lum}$ value could trigger the enantioselective polymerization of diacetylene with great

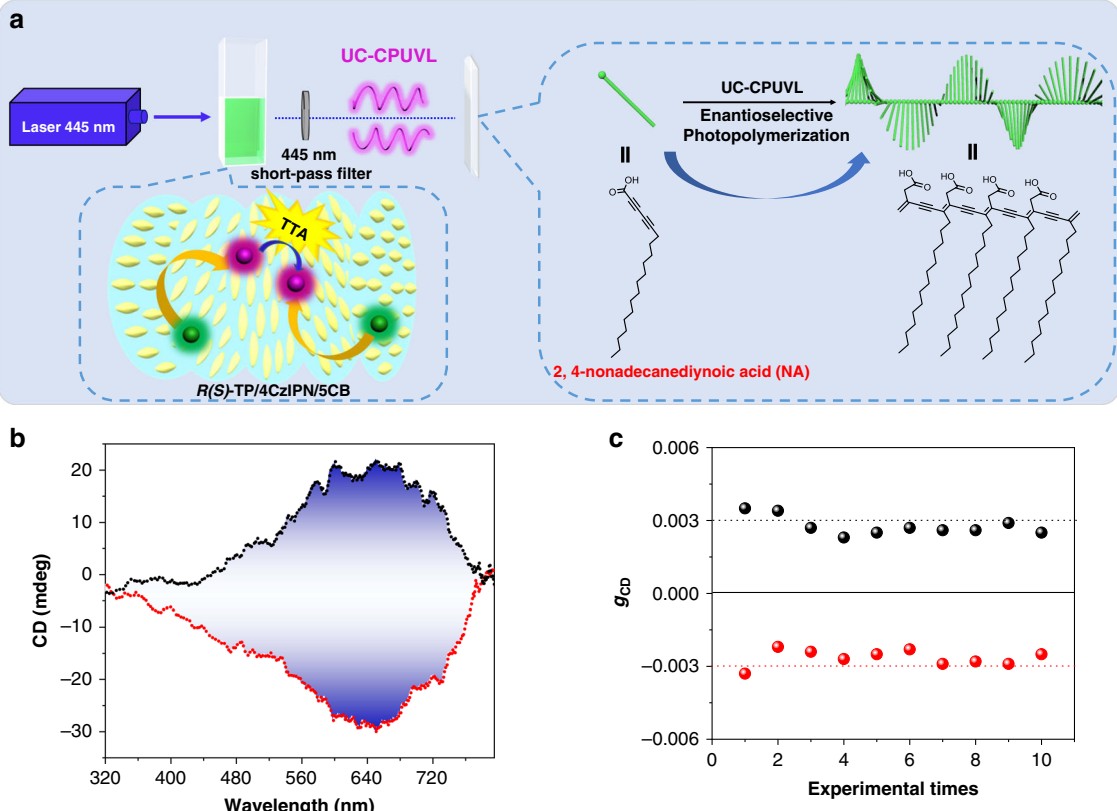

**Fig. 6 UC-CPUVL triggered enantioselective photopolymerization. a** Schematic setup of the enantioselective photopolymerization of NA. All of them were kept in the dark. A 445 nm CW semiconductor laser was used as light source. The complexes of R-TP/4CzIPN/5CB ([R-TP]/[4CzIPN] = 30/1, R-TP/5CB is 10 wt%) were sealed in a quartz cell (0.1 cm optical length) refilled with argon. The quartz plate covered with 2, 4-nonadecanediynoic acid spin-coating film was placed behind the sample, with a 445 nm short-pass filter placing between them. **b** CD spectra of PDA films after exposing to the UC-CPUVL generated from the excited R-TP/4CzIPN (black dash line) and S-TP/4CzIPN (red dash line) in N*LC with incident excitation of the 445 nm CW laser (A 445 nm short-pass was used). **c** Absorption dissymmetry factor $g_{CD}$ (640 nm) of the obtained chiral PDA in 10 different batches. The black spheres represented the $g_{CD}$ of PDA after exposing to the UC-CPUVL generated from the excited R-TP/4CzIPN/5CB and the red spheres represented the $g_{CD}$ of PDA after exposing to the UC-CPUVL generated from the excited S-TP/4CzIPN/5CB, upon excitation with 445 nm CW laser. At least three irradiation experiments were performed to obtain the average CD signals.

stereoselectivity? In order to advance this idea, we have constructed a simple irradiation platform, as shown in Fig. 6a. We selected 445 nm CW semiconductor laser as light source. According to the above demonstration, by fixing the mixing molar ratio of R-TP/4CzIPN at 30/1, an optimized mixing mass ratio of R-TP/5CB at 10 wt% was used for generating UC-CPUVL emission. Encapsulating the complexes R-TP/4CzIPN/5CB into a quartz cell (0.1 cm optical length) refilled with argon, intense UC-CPUVL was generated under shining with 445 nm CW laser. The quartz plate covered with 2, 4-nonadecanediynoic acid (NA) spin-coating film was placed behind the sample, with a 445 nm short-pass filter placing between them. After exposing to the UC-CPUVL for a few minutes, the NA film turned blue, indicating the formation of PDA. It was pleasant that the obtained PDA film showed a mirror-image Cotton effect in the CD spectra after irradiated by the UC-CPUVL with different handedness (Fig. 6b). The PDA film exhibited negative CD signal after exposing to the UC-CPUVL generated from the S-TP/4CzIPN/5CB complexes, while the positive CD signal could be achieved in the other combination. The reliability of CD spectra acquired from PDA films must be identified by testing the linear dichroism (LD), which excluded the possible effect of LD (Supplementary Fig. 11). It should be emphasized that the CD signals of PDA were in agreement with the molecular chirality of the annihilator R(S)-TP. To examine the reliability of the enantioselectivity, we have repeated the photopolymerization of NA with ten different

batches. Statistical analysis of the CD signals at 640 nm showed that the chirality of the obtained PDA always follow the molecular chirality of the annihilator (Fig. 6c). These results indicated that, after exposing to UC-CPUVL, PDA could be effectively fabricated by the enantioselective polymerization. More importantly, the chirality of PDA could be regulated by the handedness of UC-CPUVL and performed good stability and repeatability.

## Discussion

It is a very significant attempt to apply the UC-CPUVL to enantioselective photopolymerization reaction. By integrating visible-to-UV TTA-UC, chiral liquid crystal and circularly polarized UV emission, here, we proposed an effective approach for achieving UC-CPUVL with larger circular polarization. Two synthesized chiral UV emitters were used as the TTA-UC annihilators to construct the TTA-UC system, sensitized by a TADF compound 4CzIPN. Both visible-to-UV photon upconversion with a relatively high upconversion quantum efficiency ($\Phi_{UC}$ = 7.9%) and UC-CPUVL were realized. When chiral annihilators were embedded into nematic liquid crystal, the induced N*LC has endowed the upconversion system with bright UC-CPUVL activity, enabling a large $g_{lum}$ value up to 0.19. More importantly, the UC-CPUVL emission, generated from the TTA-UC process in N*LC, can trigger the enantioselective polymerization of diacetylene and perform good stability and repeatability. This work will not only provide the proof-of-concept for the usage of TTA-

UC for challenging chiral polymerization employing UC-CPUVL and paves the way for the further development of functional application of CPL-active materials.

## Methods

**Materials**. All reagents and solvents were used as received otherwise indicated. *R* (*S*)-4,12-dibromine[2.2]paracyclophane was purchased from Daicel Chiral Technologies (China) Co., Ltd. and used as received. 1,4-biphenylboronic acid was purchased from TCI and used as received. The *R*(*S*)-4,12-biphenyl[2,2]para-cyclophane (*R*-/*S*-TP) are synthesized according to literature and purified by column chromatography and confirmed the molecular structures by proton nuclear magnetic resonance (¹H NMR), matrix-assisted laser desorption–ionization-time of flight–mass spectrometry. The commercial room-temperature nematic liquid crystal, 5CB, was bought from the Chengzhi Yonghua Display Material Co., Ltd. 2,4-Nonadecadiynoic Acid was purchased from TCI and purified by dissolution in toluene and subsequent filtration to remove polymer before use.

**Characterization**. The ¹H NMR spectra were recorded on a Bruker Avance III 400 HD spectrometer. Mass spectral data were obtained by using a SolariX maldi-FTMS instrument. UV–vis spectra were recorded on Hitachi U-3900 spectro-photometer. Fluorescence spectra were measured were obtained using and F-4500 fluorescence spectrophotometer. CD and CPL spectra were measured on JASCO J-1500 and JASCO CPL-200 spectrophotometers, respectively. The fluorescence lifetime measurements were recorded on the Edinburg FLS-980 fluorescence spectrometer using time-correlated single photon counting, phosphorescence decays and upconverted emission decays were recorded on Edinburg FLS-980 using Multi-Channel Scaling. Upconverted emission spectra were recorded on a Zolix Omin-λ500i monochromator with photomultiplier tube PMTH-R 928 using an external excitation source, 445 nm semiconductor laser (Changchun New Industries Optoelectronics Tech Co., Ltd., MDL-III-445-1W), 445 nm short-pass filter (Changchun New Industries Optoelectronics Tech Co., Ltd.). POM images were recorded on a Leica DM2700M upright materials microscope. UC-CPUVL were recorded on JASCO CPL-200 spectrophotometer with an external excitation source of linearly polarized 445 nm semiconductor laser (Changchun New Industries Optoelectronics Tech Co., Ltd., MDL-III-445-1W).

## Data availability

Data supporting the findings of this study are available within the paper and its Supplementary Information files. The source data underlying Figs. 2a–c, 3a, d–f, 4a, b, 5b, and 6b are provided as a Source Data file. The data that support the findings of this study are available from the corresponding author on request. Source data are provided with this paper.

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

## Acknowledgements

This work was supported by the Strategic Priority Research Program of Chinese Academy of Sciences (No. XDB36000000); the Ministry of Science and Technology of China (2016YFA0203400 and 2017YFA0206600), National Natural Science Foundation of China (51673050 and 91856115).

## Author contributions

D.H. and X.Y. synthesized the samples and carried out all the characterizations. T.J. and P.D. supervised the work. J.H. and J.Z. helped the characterization of the nanostructure and spectral measurements. P.D. analyzed the data and wrote the paper. D.H., X.Y., and P.D. carried out the additional experiments and revised the paper. All the authors participated in the discussion and commented on the paper.

## Competing interests

The authors declare no competing interests.
