## [Peer Review File · Nature Communications]

REVIEWER COMMENTS

Reviewer #1 (Remarks to the Author):

The article entitled "Sequentially amplified circularly polarized ultraviolet luminescence for enantioselective photopolymerization" by Duan and co-workers demonstrates the construction and application of upconverted circularly polarized UV luminescence. The UC-CPUVL was obtained by integrating sensitized triplet-triplet annihilation upconversion and circularly polarized luminescence. It is impressive that the luminescence dissymmetry factor was sequentially amplified by three order of magnitude from molecular state to UC-CPUVL in chiral liquid crystal. The experimental complexity is certainly to be commended and the variety of effects and materials that are chained is impressive. At the same time, the many conceptual layers underpinning the realization of such an experiment may be interesting for a broad scope. The work looks interesting and novel, providing a significant contribution to the field of novel CPL emitters, which is undoubtedly a rapidly developing field, given the multifarious technological application it allows. This is an interesting research and could be published after following issues to be addressed.

1. How about the thickness of the material? This is a very important information, considering that in the present system, CPL mainly stems from a circular Bragg phenomenon thanks to the N*LC. Such phenomenon depends extensively on the thickness of the LC layer.
2. The experimental details of the upconversion quantum yield of TTA-UC should be added in the main text.
3. In Figure 4, the author provided the POM image of R-TP/5CB. The typical fingerprint texture proved that the N*LC has been induced perfectly by the chiral annihilator. However, in the three components system R-TP/4CzIPN/5CB, what kind of texture can be obtained? Some other evidences should be provided.
4. 445 nm laser was in this experiment. Possibly, polarized or depolarized? Again, regarding the 445 nm laser should show a provider/manufacture along with a brief specification (CW, pulse, polarized or depolarized).
5. The author should explore the effect of the optical length of a quartz cell that filled with the complexes R-TP/4CzIPN/5CB on the amplification of g values of induced PDA films.
6. To illustrate whether the observed CD signals of induced PDA films are inherent or artifact due to so-called linear dichroism (LD) in Figure 6b, the author should confirm no marked changes in the CD spectra (sign and shape) as a function of rotating angle of specimens.

Reviewer #2 (Remarks to the Author):

Han et al. report circularly polarized upconverted UV emission under visible excitation and its application in enantioselective polymerization. Based on the reported sensitizer/emitter combination of 4CzIPN and TP, they developed the chiral TTA-UC system by integrating emitter TP into chiral cyclophane building unit. They achieved a very high g value especially when the chromophores are dispersed into a nematic liquid crystal. It is also remarkable that they demonstrate the application of chiral visible-to-UV TTA-UC in the synthesis of chiral polydiacetylene. This work demonstrates the method to achieve a UV emission with a high g value and its unique application in photopolymerization. This reviewer supports the publication of this work after addressing following issues.

1. How the chromophore aggregation in nematic liquid crystal affect the g value? Since the highest g value was achieved in the nematic liquid crystal, the origin of such a high g value should be discussed more in detail.
2. The TTA-UC quantum yield and I_{th} in nematic liquid crystals should be characterized and shown.

Reviewer #3 (Remarks to the Author):

In their manuscript, the authors describe a cascade process in which a UV TTA-UC system is made chiral by incorporation into a paracyclophane scaffold, which in turn is incorporated into a liquid crystal to generate a strongly circularly polarized light. This emitted chiral light is then used to induce enantioselective polymerization of a diacetylene derivative.

Overall, the work is highly interesting in the field of CPL activity and its effective use in photochemical enantioselective processes. The idea is very elegant and the work well performed. However, I cannot recommend publication at the present stage because substantial improvements should be performed.

- The English writing should be significantly improved. Indeed, there are a few sentences that are difficult to understand due to incorrect English language.
- Page 7 : « from the chiral molecules R(S)-4,12-biphenyl[2,2]paracyclophane and 1,4-biphenylboronic acid ». It should be from the bromo derivative.
- Figure 2 : legends precisising to which stereoisomer each colored spectrum corresponds should be added.
- Page 8 : what is « a Cotton splits » ?
- The experimental set-ups of non commercial apparatus should be described in the SI file.
- Page 13 or 14 : « the intensity of TTA-UC emission was relatively weak ». How much ? Please quantify. What is the impact on the polymerization process ? Why is the process sufficiently effective ?
- Page 17 : the « discussion » should be the « conclusion »

Reviewer: 1

Comments:

The article entitled “Sequentially amplified circularly polarized ultraviolet luminescence for enantioselective photopolymerization” by Duan and co-workers demonstrates the construction and application of upconverted circularly polarized UV luminescence. The UC-CPUVL was obtained by integrating sensitized triplet-triplet annihilation upconversion and circularly polarized luminescence. It is impressive that the luminescence dissymmetry factor was sequentially amplified by three orders of magnitude from molecular state to UC-CPUVL in chiral liquid crystal. The experimental complexity is certainly to be commended and the variety of effects and materials that are chained is impressive. At the same time, the many conceptual layers underpinning the realization of such an experiment may be interesting for a broad scope. The work looks interesting and novel, providing a significant contribution to the field of novel CPL emitters, which is undoubtedly a rapidly developing field, given the multifarious technological application it allows. This is an interesting research and could be published after following issues to be addressed.

Response: Thank you very much for your support and encouragement.

(1) How about the thickness of the material? This is a very important information, considering that in the present system, CPL mainly stems from a circular Bragg phenomenon thanks to the N*LC. Such phenomenon depends extensively on the thickness of the LC layer.

Response: Thanks for your detailed reading and important comment. The thickness of the chiral liquid crystal layer in the present system is 0.1 mm. It should be emphasized that the Bragg reflection of N*LC is consistent with its photonic band gap. Unfortunately, because the weak helical twist power (HTP) of chiral emitter, the photonic band gap of N*LC could not be regulated in the visible light region. Thus, the Bragg phenomenon could be neglected on this work. Nevertheless, the effect of Bragg phenomenon on the UC-CPUVL is a valuable suggestion, encouraging us to explore in future works.

(2) The experimental details of the upconversion quantum yield of TTA-UC should be added in

the main text.

Response: Thanks for your comment. Firstly, we carefully measured the upconversion emission of *R*-TP/4CzIPN with different incident excitation intensities of the 445 nm CW laser in deaerated toluene solution (Fig. R1a). Then the luminescence intensity of Coumarin 6 exhibited linear increase as the excitation power increased (Fig. R1b and 1c), suggesting a constant quantum yield independent of excitation power density. The Φ_{UC} can be calculated by comparing upconversion luminescence with the Coumarin 6 in the DMF solution under identical test conditions. The upconverted emission quantum yield (Φ_{UC}) was determined relative to a standard according to the following equation

$$\Phi_{UC} = 2\Phi_{std} \left(\frac{A_{std}}{A_{uc}} \right) \left(\frac{I_{uc}}{I_{std}} \right) \left(\frac{\eta_{uc}}{\eta_{std}} \right)^2$$

Where Φ , A , I and η represent a quantum yield, absorbance, integrated photoluminescence spectral profile, and refractive index of the solvents used as a standard, respectively. The subscripts *UC* and *std* denote the parameters of the tested upconversion and standard systems, respectively. The UC quantum yield was determined relative to a standard, Coumarin 6 in DMF ($\Phi_{std} = 72\%$) under 445 nm excitation. Note that the theoretical maximum of Φ_{UC} is standardized to be 1 (100%). Finally, the upconversion quantum yield of TTA-UC was calculated (Fig. R1d). We rewrote the corresponding parts in our revised manuscript.

Figure. R1. a) Upconversion emission spectra of *R*-TP/4CzIPN with different incident excitation intensities of the 445 nm CW laser in deaerated toluene solution; b) Fluorescence spectra of Coumarin 6 in DMF with different incident excitation intensities of the 445 nm CW laser; c) Linear fitting of the fluorescence intensity of Coumarin 6 in DMF as a function of the excitation intensity; d) Upconversion quantum yield of *R*-TP/4CzIPN in deaerated toluene solution with different excitation light intensities of the 445 nm CW laser.

(3) In Figure 4, the author provided the POM image of *R*-TP/5CB. The typical fingerprint texture proved that the N*LC has been induced perfectly by the chiral annihilator. However, in the three components system *R*-TP/4CzIPN/5CB, what kind of texture can be obtained? Some other evidences should be provided.

Response: Thanks for the reminding. After embedding with 4CzIPN, the POM image of three components system *R*-TP/4CzIPN/5CB was shown in Fig. R2 or Supplementary Fig. 6. The fingerprint texture could still be preserved, which indicates the typical fingerprint texture of N*LC could not be destroyed by the sensitizer 4CzIPN. The POM image of *R*-TP/4CzIPN/5CB was added in Supplementary Information. In addition, we rewrote the corresponding parts in the present revision.

Figure. R2 or Supplementary Figure 6. POM image of three components system *R*-TP/4CzIPN/5CB (*R*-TP/5CB = 10 wt%, 4CzIPN/*R*-TP = 3 mol%).

(4) 445 nm laser was in this experiment. Possibly, polarized or depolarized? Again, regarding the 445 nm laser should show a provider/manufacture along with a brief specification (CW, pulse, polarized or depolarized).

Response: Thanks for your comment. The linearly polarized 445 nm semiconductor laser was used in this work, which is the same with all the normal semiconductor laser. We guess the reviewer considered the influence of 445 nm laser to the UC-CPUVL signals. Due to the signal of UC-CPUVL was determined by the N*LC, linearly polarized 445 nm laser has no effect on the UC-CPUVL signals. The 445 nm semiconductor laser was bought from Changchun New Industries Optoelectronics Technology Co., Ltd.. Linearly polarized 445 nm laser with MDL-III-445-1W model was used in this work. The information of 445 nm semiconductor laser was added in the Characterizations.

(5) The author should explore the effect of the optical length of a quartz cell that filled with the complexes *R*-TP/4CzIPN/5CB on the amplification of g values of induced PDA films.

Response: Thank you very much for your comment. The thickness of the chiral liquid crystal layer in the present system is 0.1 mm. It should be emphasized that the Bragg reflection of N*LC is consistent with its photonic band gap. Unfortunately, because the weak helical twist power (HTP) of chiral emitter, the photonic band gap of N*LC could not be regulated in the visible light region. Thus, the Bragg phenomenon could be neglected on this work. However, when increasing the thickness of LC, the intensity of UC-CPUVL will be suppressed due to the strong reabsorption, while the g_{lum} value will not be influenced seriously. Thus, the g_{CD} values of induced PDA films

will not be influenced by the optical length of a quartz cell because the UC-CPUVL g_{lum} value about 0.2 is unchanged.

(6) To illustrate whether the observed CD signals of induced PDA films are inherent or artifact due to so-called linear dichroism (LD) in Figure 6b, the author should confirm no marked changes in the CD spectra (sign and shape) as a function of rotating angle of specimens.

Response: Thank you very much for your valuable suggestions. In order to answer to this question, it would be a good idea to investigate the CD spectra by rotating angle of specimens. To exclude the possible effect of linear dichroism, CD measurements were carried out by rotating the sample about the normal of the film. The variation of CD signals at all rotation angles was small, indicating that the main origin of CD should be the helical chains of PDA, as shown in Fig. R3 or Supplementary Fig. 10. Corresponding parts were added in Supplementary Information.

Figure. R3 or Supplementary Figure 10. The CD spectra at various rotation angles about surface normal for the PDA film after exposing to the UC-CPUVL generated from the excited *R*-TP/4CzIPN in N*LC with incident excitation of the 445 nm CW laser (A 445 nm short-pass was used).

Reviewer: 2

Comments:

Han et al. report circularly polarized upconverted UV emission under visible excitation and its application in enantioselective polymerization. Based on the reported sensitizer/emitter combination of 4CzIPN and TP, they developed the chiral TTA-UC system by integrating emitter TP into chiral cyclophane building unit. They achieved a very high g value especially when the chromophores are dispersed into a nematic liquid crystal. It is also remarkable that they demonstrate the application of chiral visible-to-UV TTA-UC in the synthesis of chiral polydiacetylene. This work demonstrates the method to achieve a UV emission with a high g value and its unique application in photopolymerization. This reviewer supports the publication of this work after addressing following issues.

Response: Thank you very much for your support. We appreciate these highly encouraging comments, as well as constructive suggestions.

(1) How the chromophore aggregation in nematic liquid crystal affect the g value? Since the highest g value was achieved in the nematic liquid crystal, the origin of such a high g value should be discussed more in detail.

Response: Thanks for this comment. The chiral chromophores used in this work can molecularly disperse in host liquid crystal without obvious aggregation. After dispersing chiral chromophores into the nematic liquid crystal, the g_{lum} value of CPL emission is determined by the chirality of chiral nematic liquid crystal. Normally, in a certain range, increasing the weight ratio of chiral molecule is favorable to the induction of chiral nematic liquid crystal. However, the properties of host liquid crystal (room-temperature nematic liquid crystal 5CB is used in this work) will perform significant changes, such as viscosity increase or clear point decrease, after adding an excess of chiral dopants. Thus, it is essential to thoroughly explore the CPL activity of various mixing weight ratios of chiral dopant. As shown in Fig. 4a, the highest g_{lum} value is obtained when the mixing weight ratio of *R*-TP/5CB was 10 wt%, indicating that the chiral nematic liquid crystal possessed the most favorable configuration in this case. The corresponding part is rewritten in the revised version.

(2) The TTA-UC quantum yield and I_{th} in nematic liquid crystals should be characterized and shown.

Response: Thank you very much for your suggestion. We have added the corresponding experimental results in the revised manuscript. In the excitation of 445 nm CW laser, the upconverted UV emission spectra of *R*-TP/4CzIPN in N*LC were obtained (Fig. R4a), whose corresponding double-logarithmic plots were showed in Fig. R4b. The black and red lines are the fitting results with slopes of 1.9 and 1.0 in the low and high excitation intensity ranges, respectively. It involves the TTA process. The threshold excitation intensity (I_{th}) is 270 mW cm^{-2} , where the UC quantum yield reached saturation. In addition, the saturated TTA-UC quantum efficiency Φ_{UC} in liquid crystal was about 2.3%, which was obtained for *R*-TP/4CzIPN in 5CB by using Coumarin 6 in 5CB as a standard and with the theoretical maximum of 100% (Fig. R4c). Corresponding parts were added in Supplementary Information.

Figure. R4 or Supplementary Figure 9. a) Upconversion emission spectra of *R*-TP/4CzIPN with different incident excitation intensities of the 445 nm CW laser in a room-temperature nematic liquid crystal (5CB); b) Double-logarithmic plots of the upconversion emission intensity of *R*-TP/4CzIPN in 5CB as a function of the excitation intensity; c) Upconversion efficiencies of *R*-TP/4CzIPN in 5CB with different excitation light intensities of the 445 nm CW laser. For all the upconversion measurements, a 445 nm short-pass filter was used.

Reviewer: 3

Comments:

In their manuscript, the authors describe a cascade process in which a UV TTA-UC system is made chiral by incorporation into a paracyclophane scaffold, which in turn is incorporated into a liquid crystal to generate a strongly circularly polarized light. This emitted chiral light is then used to induce enantioselective polymerization of a diacetylene derivative. Overall, the work is highly interesting in the field of CPL activity and its effective use in photochemical enantioselective processes. The idea is very elegant and the work well performed. However, I cannot recommend publication at the present stage because substantial improvements should be performed.

Response: Thank you very much for your support. We appreciate these highly encouraging comments, as well as constructive suggestions.

(1) The English writing should be significantly improved. Indeed, there are a few sentences that are difficult to understand due to incorrect English language.

Response: Thanks for this comment. We have thoroughly checked the manuscript and polished some expressions in revised version to make them easier to understand.

(2) Page 7: «from the chiral molecules *R(S)*-4,12-biphenyl[2,2]paracyclophane and 1,4-biphenylboronic acid». It should be from the bromo derivative.

Response: Thanks for this valuable suggestion. It should be “*R(S)*-4,12-dibromo[2,2]paracyclophane”, and the corresponding parts have been corrected in the revised version.

(3) Figure 2: legends precisising to which stereoisomer each colored spectrum corresponds should be added.

Response: Thanks for this important comment. The legends were added and corresponding parts have been corrected in the revised manuscript.

(4) Page 8: what is « a Cotton splits »?

Response: Thanks for your professional recommendation. It should be ‘Cotton effect’, which is used to describe the property of optical rotation. The positive CD signal is consistent with the positive ‘Cotton effect’ while the negative signal represents the opposite one. The corresponding parts have been corrected in the revised manuscript.

(5) The experimental set-ups of non commercial apparatus should be described in the SI file.

Response: Thanks very much for your thoughtful suggestion. In this work, the upconverted emission spectra (Fig. 3a, Supplementary Fig. 4a and Supplementary Fig. 9) were conducted by the home-build installation, as shown in Fig. R5, which contained the follow components: 445 nm semiconductor laser (Changchun New Industries Optoelectronics Tech Co., Ltd, MDL-III-445-1W), 445nm short-pass filter (Changchun New Industries Optoelectronics Tech Co., Ltd.) and Zolix Omin- λ 500i monochromator with photomultiplier tube PMTH-R 928. Corresponding part was added in Supplementary Information.

Figure. R5 or Supplementary Scheme 1. Schematic illustration of the experimental setup for characterization of TTA-UC emission.

(6) Page 13 or 14: « the intensity of TTA-UC emission was relatively weak ». How much? Please quantify. What is the impact on the polymerization process? Why is the process sufficiently effective?

Response: Thank you very much for your suggestion. According to your suggestion, we have

evaluated the TTA-UC efficiency by a relative method using Coumarin 6 in liquid crystal as a reference. Comparing with the solution state ($\Phi_{UC} = 7.9\%$), the intensity of TTA-UC emission in liquid crystal was suppressed ($\Phi_{UC} = 2.3\%$). However, this upconversion efficiency is strong enough to initiate the photopolymerization of diacetylene. In this work, the used monomer 2, 4-nonadecanedioic acid (NA) film turning blue indicates the formation of polydiacetylene (PDA) when excited by UC-CPUVL. Thus, it clearly demonstrate that the intensity of TTA-UC emission in N*LC is enough to initiate enantioselective photopolymerization of diacetylene and the process was sufficiently effective. On the other hand, the UC-CPUVL emission with a high g_{lum} (0.2) can effectively trigger the enantioselective polymerization of diacetylene. The chirality of PDA could be regulated by the handedness of UC-CPUVL and perform good stability and repeatability.

(7) Page 17: the « discussion » should be the « conclusion ».

Response: Thank you very much for the reminding. We have corrected it in the revised manuscript.

REVIEWERS' COMMENTS

Reviewer #1 (Remarks to the Author):

The author answered all my questions very well, and I recommended that the manuscript could be published.

Reviewer #2 (Remarks to the Author):

The authors properly addressed my comments. This reviewer supports the publication of the revised manuscript.

Reviewer #3 (Remarks to the Author):

The authors have taken into account all the comments of the three reviewers and made their manuscript even more convincing and their scientific results even stronger.

I therefore strongly recommend publication in Nature Communications journal.

Reviewer: 1

Comments:

The author answered all my questions very well, and I recommended that the manuscript could be published.

Response: Thank you very much for your support and encouragement.

Reviewer: 2

Comments:

The authors properly addressed my comments. This reviewer supports the publication of the revised manuscript.

Response: Thank you very much for your support and encouragement.

Reviewer: 3

Comments:

The authors have taken into account all the comments of the three reviewers and made their manuscript even more convincing and their scientific results even stronger.

I therefore strongly recommend publication in Nature Communications journal.

Response: Thank you very much for your support and encouragement.